# Novel Ultrasound Examination and Guided Intervention of Peri-Oral Musculature and Fascia in Wind Players with Embouchure Problems: Technical Note

**DOI:** 10.3390/diagnostics15050514

**Published:** 2025-02-20

**Authors:** Daniel Chiung-Jui Su, Mei-Chen Yeh, King Hei Stanley Lam

**Affiliations:** 1Department of Physical Medicine and Rehabilitation, Chi Mei Medical Center, Tainan 710, Taiwan; dr.daniel@gmail.com; 2A Tempo Regeneration Center for Musicians, Tainan 700, Taiwan; 3Department of Endocrinology, Chi Mei Medical Center, Tainan 710, Taiwan; jamie.wien@gmail.com; 4Faculty of Medicine, The Chinese University of Hong Kong, Hong Kong, China; 5Faculty of Medicine, The University of Hong Kong, Hong Kong, China; 6Board of Clinical Research, The Hong Kong Institute of Musculoskeletal Medicine, Hong Kong, China; 7Center for Regional Anesthesia and Pain Medicine, Wan Fang Hospital, Taipei Medical University, Taipei 110, Taiwan; 8Center for Regional Anesthesia and Pain Medicine, Chung Shan Medical University Hospital, Taichung 402, Taiwan

**Keywords:** embouchure dysfunction, wind player, ultrasound, peri-oral musculature, fascia, sonoanatomy, ultrasound-guided intervention

## Abstract

**Background**: Embouchure problems in wind players can severely affect musical performance. The complexity of the embouchure involves peri-oral musculature, which is essential for controlling airflow and tone production but is prone to injuries from overuse or misuse. The current literature lacks guidance on utilizing ultrasound for diagnosing embouchure-related injuries. **Methods**: This technical note presents a scanning method for wind players that presented with embouchure problems, with common pathological sonographic findings and ultrasound-guided interventions. **Results**: A comprehensive overview of the sonoanatomy of the peri-oral musculature relevant to the embouchure and a scanning protocol for the embouchure in wind players are described. This technical note also highlights common pathological sonographic findings associated with embouchure problems and describes ultrasound-guided interventions addressing these issues. **Conclusions**: This technical note emphasizes the potential of ultrasound in diagnosing and treating embouchure problems, contributing to effective therapeutic strategies for wind players.

## 1. Introduction

Ultrasound has been employed in the facial area for nerve block or botulinum toxin injections; however, the literature is limited regarding its application to the embouchure in wind players [1,2]. The embouchure encompasses the efforts required to control the amount, pressure, and direction of airflow by adjusting the tension of the peri-oral and fascial muscles and fasciae, as well as the positioning and movement of the tongue, jaws, and cheeks to produce the desired tone on the wind instrument [3]. The tension in the embouchure is continuously maintained and adjusted by the muscles and fasciae surrounding the orbicularis oris muscle, which includes the orbicularis oris itself, levator anguli oris (LAO), zygomaticus major (ZMa) and zygomaticus minor (ZMi), buccinator, risorius, depressor anguli oris (DAO), and the modiolus [3]. Like any skeletal muscle, embouchure muscles can sustain injuries, ranging from Grade I to Grade III, or may develop scar-like fibrotic tissue within the muscle. The compromised integrity of these muscles and the associated fascia, which can be detected through sonopalpation [4,5], may hinder the embouchure’s ability to counteract intra-oral pressure (IOP), thereby impairing the player’s control over the embouchure.

Wind players with embouchure problems may present symptoms such as inability to form the required embouchure, soreness with tenderness, tremor of the affected muscles, air leakage from the mouthpiece, increased intra-oral space with a puffed mouth when playing, tenderness, and poor endurance [3,6]. Their performance may be compromised due to their inability to play in specific or all registers, failure to perform vibrato or hold a long note, poor articulation, and poor tone formation.

Additionally, embouchure dystonia, associated with plastic maladaptive changes in sensorimotor networks, is often preceded by embouchure problems [3]. Therefore, if local injuries could be treated and healed, the feedback from the local tissues could be restored and help promote recovery from the embouchure dystonia.

Embouchure problems in wind players can severely affect musical performance. The complexity of the embouchure involves peri-oral musculature, which is essential for controlling airflow and tone production but is prone to injuries from overuse or misuse. The current literature primarily utilizes surface electromyography to assess the function of embouchure muscles. However, distinguishing individual muscle activity remains challenging due to the close proximity of these muscles [7]. Additionally, there is a lack of the literature exploring the use of ultrasound for diagnosing embouchure-related injuries.

This technical note aims to achieve three key objectives: first, to provide a comprehensive overview of the sonoanatomy of the peri-oral musculature relevant to the embouchure; second, to present a scanning protocol for the embouchure in wind players and to highlight common pathological sonographic findings associated with embouchure problems; and third, to detail ultrasound-guided interventions that may assist in the diagnosis and treatment of these issues.

## 2. Materials and Methods

### 2.1. Sonoanatomy of Peri-Oral Musculature and Fascia System Concerning Embouchure

The integrity of the peri-oral fascia is crucial for forming the embouchure. The most superficial layer underneath the skin in the peri-oral region is the superficial musculoaponeurotic system (SMAS). It is a fibro-membranous structure consisting of muscle fibers, collagen fibers, and elastic fibers with a density similar to muscles. The SMAS acts as the dermal termination of the peri-oral musculature, including the platysma muscle, and is tensed by them [8]. Underneath the SMAS is the peri-oral musculature. The orbicularis oris (OO) is a sphincter muscle that is closely interdigitated with adjacent muscles. It can be divided into the superior and inferior portions by the oral commissure. These two portions can be divided into the inner marginal and outer labial parts [9]. The inner marginal part of the OO is important in embouchure formation because it directly connects with the buccinator.

Lateral to the OO, the modiolus can be found at the lateral border of the mouth’s corner. The modiolus is the central hub connecting the pulling force from the LAO, ZMa, risorius, buccinator, and DAO (Figure 1). Therefore, we can use the modiolus as the reference point during scanning. Histological studies have shown that the modiolus consists of tendinous structures with dense, irregular collagenous connective tissue, making it an isoechoic to hyperechoic structure under ultrasound [10]. This fibromuscular tendinous structure located at the lateral border of the mouth is intermingled with muscle fibers and the superficial fascia of the face. As such, the location of the modiolus can be affected by the tension around it in the long term. In the average population, a downward displacement may occur due to aging; however, in musicians, their playing technique influences the position in the long run. The asymmetry of the modiolus position between the left and right sides of the face can often be seen in musicians who do not place their mouthpiece in the midline of the OO, which may be habitual or due to underlying tooth inclination.

The levator anguli oris and depressor anguli oris act as counterforces on the modiolus, controlling its position and tension [9]. These two muscles are also actively used by reed players who place the mouthpiece between the lips.

The buccinator can be divided into the upper oblique part, the middle transverse part, and the lower oblique part, based on the origin from the maxilla, the pterygomandibular raphe, and the mandible, respectively. The lower oblique part is inserted adjacent to the DAO, while the upper oblique part is adjacent to the LAO [9]. The middle transverse part exerts a direct tractional force over the modiolus laterally, which is heavily utilized during embouchure formation and is prone to injury. The failure of the pulling tension from the buccinator can lead to the oral membrane protruding from the defect, which decreases the IOP and significantly affects articulation.

When playing a wind instrument, the OO has direct contact with the mouthpiece. For brass players, the lower half of the OO will have firm contact with the mouthpiece and needs to provide sustainable stability, primarily stabilized by the middle layer to assist the upper and lower oblique parts of the buccinator along with the modiolus [11]. Conversely, the upper half of the OO must vibrate freely to produce sound, with tension being controlled by the superficial layer. When the deep layer has some deficit due to an injury, the inner marginal part of the OO cannot withstand the IOP, causing it to flip out during playing, resulting in air leakage or attack blockage, leading to lag in the onset of notes or poor articulation (Figure 2a–d).

### 2.2. Scanning Protocol for the Embouchure in Wind Players

When scanning the embouchure in wind players, we should increase the IOP by asking the patient to play their instrument while scanning to visualize pathologies more clearly. The embouchure muscles will be stretched and tensioned to counteract the increased IOP, allowing us to observe structural and functional deficits. During the examination, we assess individual muscles and the fascia integrity of the three layers of the peri-oral structures. Additionally, if the players fail to play in a specific register, examiners should request that the patient play in that particular register for a better analysis of embouchure dysfunction.

The scanning of the embouchure started from the lateral border of the modiolus, approximately 10–20 mm lateral and 0–10 mm below the mouth corner (cheilion). A linear transducer with a frequency of no less than 18 MHz is recommended, with the focus set at a depth of 1 to 2 cm. The depth may need to be adjusted to 2 to 3 cm deep for buccinator muscles.

First, the probe is placed in a sagittal orientation, aligned with the LAO, modiolus, and DAO. The SMAS, LAO, OO, DAO, modiolus, and the upper oblique and lower oblique parts of the buccinator can be examined together in this view.

The examiner can move the transducer in a sagittal plane to assess the integrity of the LAO and DAO at their origins. The deep fascia of the platysma can be found lying on top of the origin of the DAO.

Secondly, we pivot the probe to check for the connectivity of the ZMa, ZMi, and levator labii superior to the upper region of the OO.

Thirdly, the probe is positioned in a horizontal plane aligned with the buccinator. The middle transverse, upper oblique, and lower oblique portions of the buccinator should be examined individually. This layer is also the deepest layer of the peri-oral musculature, contributing significantly to the tension in the embouchure through abduction (Figure 3a–c).

When scanning for the buccinator, it is crucial to ask the player to play while scanning to help reveal the lesions and evaluate the tensional balance (Figure 3d,e).

Lastly, both the superior and inferior OO are scanned to check for any tears, particularly over the inner marginal area. The mouthpiece is removed during the scanning of the OO to provide better access for the ultrasound probe. Additionally, the OO is relatively sensitive to tenderness compared to other embouchure structures; therefore, sonopalpation can help identify an active lesion (Figure 4).

As a reminder, the middle transverse part originates from the pterygomandibular raphe, which is covered by the mandible and is not visible through the ultrasound.

During scanning, we should identify any tears, chronic fibrotic changes, tendinopathy, tremor, or dystonic movement of the peri-oral muscles. When a full-thickness tear or complete muscle rupture occurs, ultrasound may reveal the oral membrane protruding from the defect as a hyperechoic layer when the IOP increases.

## 3. Results

In the starting position, the probe is placed in a sagittal orientation, aligned with the LAO, modiolus, and DAO. Hypoechoic changes with increased tear size when IOP increases indicate an injury over the muscles (Figure 5). We can classify the severity of muscle injury based on Chan et al. [12]: Grade I is characterized by increased echogenicity without architectural distortion, Grade II is defined by discontinuous muscle fibers with altered echogenicity, and Grade III presents as the complete discontinuity of muscle fibers with the retraction of the muscle ends.

When the probe is moved sagittally to scan the origins of the LAO and DAO, an interruption with irregularity in the fibrillary pattern of the muscle and fascial layer indicates a Grade II tear. Additionally, comparing both sides of the embouchure can help differentiate abnormal findings from normal ones (Figure 6).

In the second step of the protocol, we scan the connectivity of the ZMa, ZMi, and levator labii superior to the upper region of the OO; a hypertrophied muscle may be the compensatory result of the long-term playing of an instrument, which should be differentiated from true lesions (Figure 7).

## 4. Discussion

For wind players, maintaining control over the embouchure requires constant and precise adjustments to its tension and shape. When the peri-oral fascia and muscles are injured, managing the embouchure becomes increasingly difficult and may result in embouchure dysfunction. This scanning protocol allows physicians to assess the embouchure-related muscles and fascia, examining their integrity during instrument playing. Furthermore, once an injured embouchure is identified, we can develop a more detailed treatment plan. The literature suggests that dry needling is effective in reducing muscle stiffness and alleviating pain in muscle injuries [13]. However, when applied to wind players with embouchure dysfunction, the results are often unsatisfactory, particularly regarding embouchure control and durability while playing.

Regenerative injection therapy is known to promote tissue healing, and ultrasound-guided injections of regenerative agents, such as high concentrations of dextrose or platelet-rich plasma, have been shown to facilitate tissue recovery [14,15,16].

According to a case series by Yeh et al., 19 wind players with embouchure injuries, with symptoms persisting for an average of 3.8 years, underwent platelet-rich plasma injections administered twice, two weeks apart. Among them, 81% reported improved embouchure control and a return to regular performance, with an average recovery time of four months. Those who did not respond well exhibited symptoms such as air leakage at the edge of the mouthpiece or dystonic movements of the embouchure or facial muscles, indicating severe embouchure dysfunction and/or embouchure dystonia [17]. In players who showed improvement, hyperechoic tissue with increased fiber continuity was observed (Figure 8).

When performing ultrasound-guided injections to embouchure structures, either an out-of-plane or in-plane approach can be used, with approximately 0.5 to 1 cc of injectate administered at each lesion. Some post-injection swelling may occur, but it is generally tolerable and subsides within a few days. Players should be advised to reduce the intensity of their playing during the recovery period to facilitate optimal healing.

Lastly, several neuromuscular structures should be visualized prior to intervention in the peri-oral embouchure to avoid puncture. First, the facial artery typically runs lateral to the modiolus, branching into the superior labial and inferior labial arteries, which supply the upper and lower parts of the OO. Second, the buccal branch of the facial nerve runs above the buccinator and gives off several branches that form the infraorbital plexus. Since some of these nerves are too small to be adequately detected with ultrasound, the needle should be inserted slowly through the peri-oral region. Additionally, a hydrodissection technique [18] may be employed, using the injectate to push small blood vessels and nerves away, thereby minimizing the risk of damage to these delicate structures.

The ultrasound scanning of the peri-oral structures can be a valuable tool in speech therapy and maxillofacial rehabilitation. By providing the real-time imaging of the muscle and soft tissue dynamics, practitioners can better assess the functional integrity of the embouchure and surrounding areas. This imaging allows for targeted interventions, enhancing the effectiveness of rehabilitation strategies. Furthermore, integrating ultrasound into therapy can help tailor individualized treatment plans, monitor progress, and adjust techniques based on visual feedback, ultimately improving outcomes for patients with embouchure dysfunction and related speech challenges.

## 5. Conclusions

Ultrasound provides a real-time assessment of the embouchure during playing. This technical report highlights the importance of ultrasound in understanding and managing embouchure-related dysfunction in wind players. The ability of physicians to diagnose and treat local muscle and fascia injuries in the embouchure region can potentially restore proper embouchure function and improve performance. This underscores the need for further research, including the addition of other scanning modalities, such as elastography, and their clinical application.

## Figures and Tables

**Figure 1 diagnostics-15-00514-f001:**
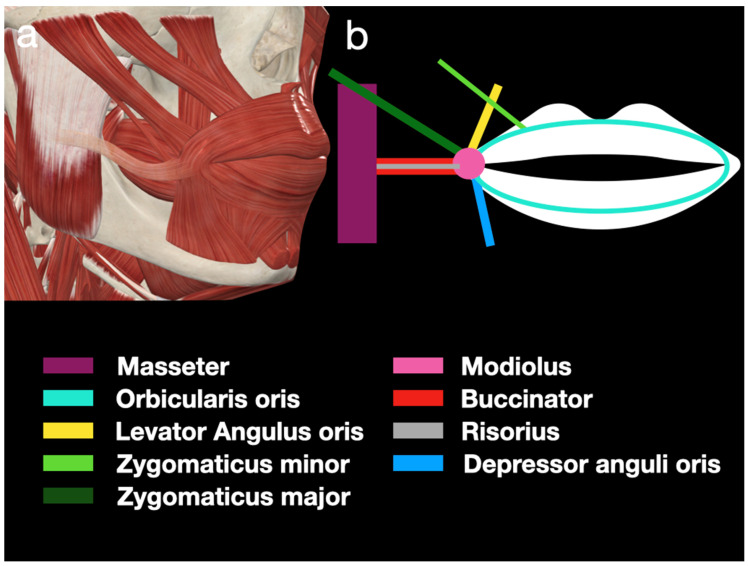
(**a**,**b**) A schematic drawing of the peri-oral musculature related to the embouchure.

**Figure 2 diagnostics-15-00514-f002:**
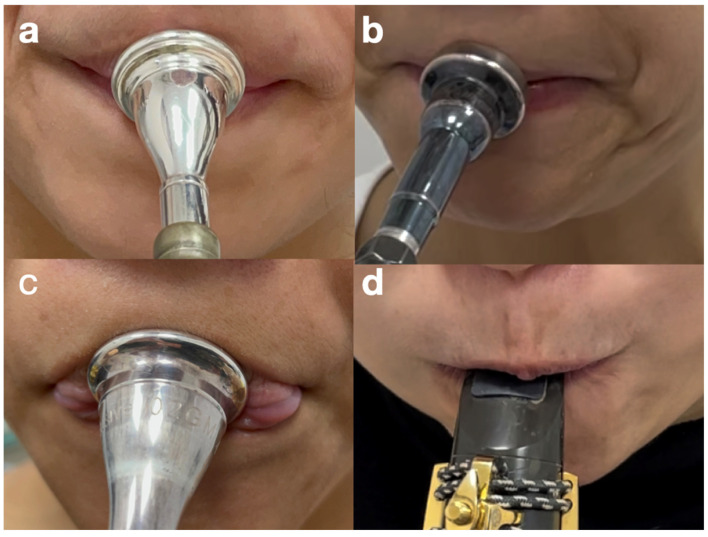
(**a**,**b**) The asymmetry of the mouth angle indicating underlying fascia imbalance related to the modiolus. (**c**) The inner marginal part of the orbicularis oris flipped out during play due to injuries of the buccinator of a horn player. (**d**) The failure of the right side of the fascial tension of the embouchure when the intra-oral pressure increased during playing.

**Figure 3 diagnostics-15-00514-f003:**
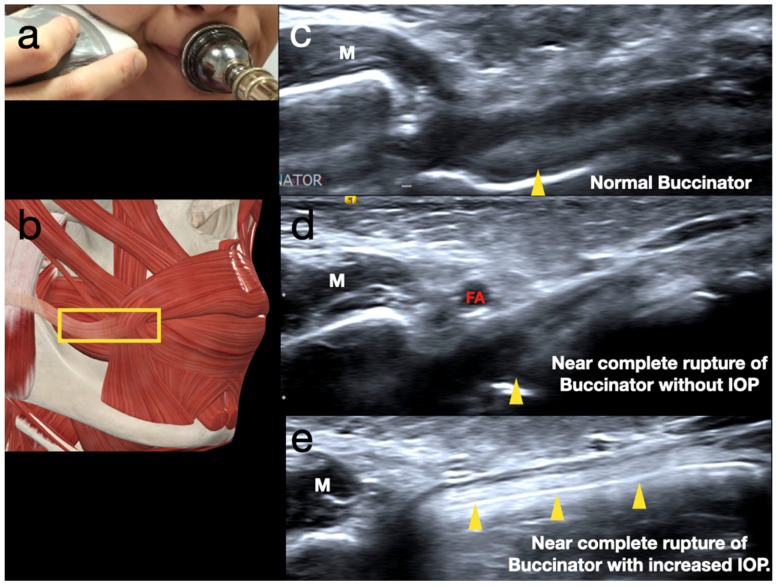
(**a**) Probe position when scanning the buccinator. (**b**) A schematic drawing of buccinator scanning. The yellow rectangle is the probe position. (**c**) The normal ultrasound appearance of the buccinator in wind players. The yellow arrowhead indicates the interface between the air and oral membrane. (**d**) The partial tear of the buccinator is difficult to visualize due to a lack of intra-oral pressure (IOP). The yellow arrowhead represents the partially raised interface between the air and the oral membrane. (**e**) Only a thin layer of the buccinator remains when scanned during playing. The buccinator fails to hold against the IOP when playing. The hyperechoic line shows the intra-oral air pushing the oral mucous against the buccinator. The yellow arrowheads mark the raised interface between the air and the oral membrane. M: masseter; FA: facial artery.

**Figure 4 diagnostics-15-00514-f004:**
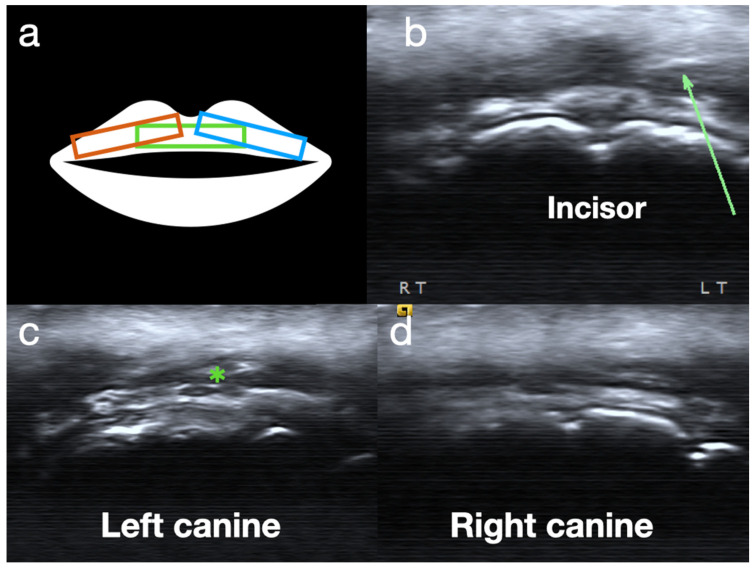
Ultrasound images of the orbicularis oris. (**a**) A schematic drawing of the probe position in (**b**–**d**). (**b**) green rectangle; (**c**) blue rectangle; (**d**) orange rectangle. (**b**) Mild fibrosis over the green arrow area. (**c**) Mild fibrosis over the green asterisk area superficial to the left canine teeth. (**d**) Normal OO superficial to the right canine teeth.

**Figure 5 diagnostics-15-00514-f005:**
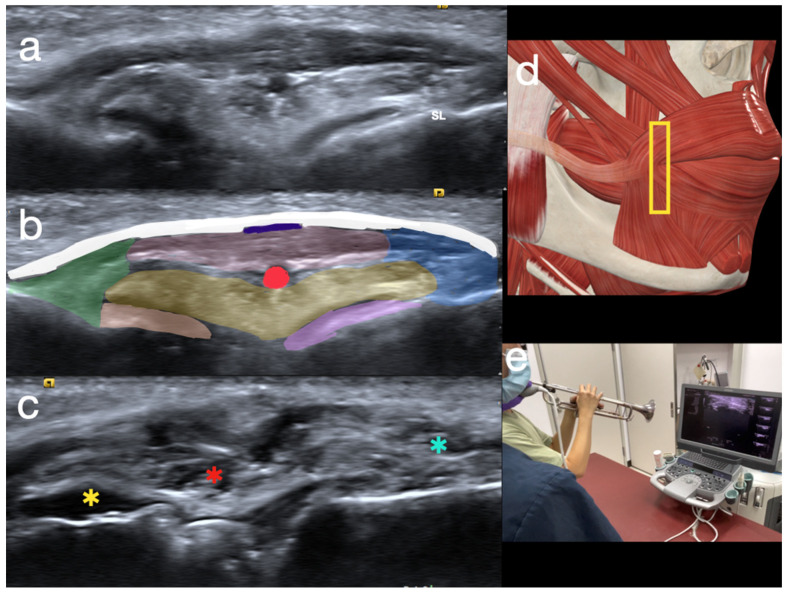
Ultrasound images of the modiolus in the transverse view. (**a**) A normal embouchure in wind players. (**b**) A schematic view of the normal embouchure (green: levator anguli oris (LAO); white: superficial musculoaponeurotic system (SMAS); red: orbicularis oris (OO); blue: depressor anguli oris (DAO); yellow: modiolus; orange: upper oblique buccinator; purple: lower oblique buccinator). (**c**) Anechoic areas over the modiolus (red asterisk), LAO (yellow asterisk), and DAO (blue asterisk) indicate partial tears. SL: superior labial artery. (**d**) Probe position (yellow rectangle) during scanning. (**e**) Scanning performed while playing an instrument to increase the IOP.

**Figure 6 diagnostics-15-00514-f006:**
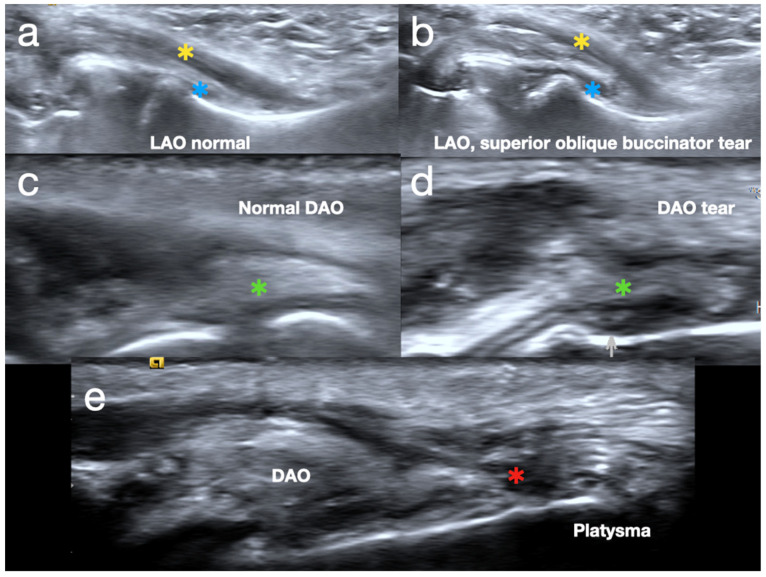
(**a**) Normal levator anguli oris (LAO) (yellow asterisk) and upper oblique buccinator (blue asterisk). (**b**) Grade I tear of LAO (yellow asterisk) and grade II tear of upper oblique buccinator (blue asterisk). (**c**) Normal depressor anguli oris (DAO) (green asterisk) (**d**) Grade II tear of DAO (green asterisk). (**e**) Tear of fascia (red asterisk) in between platysma and DAO.

**Figure 7 diagnostics-15-00514-f007:**
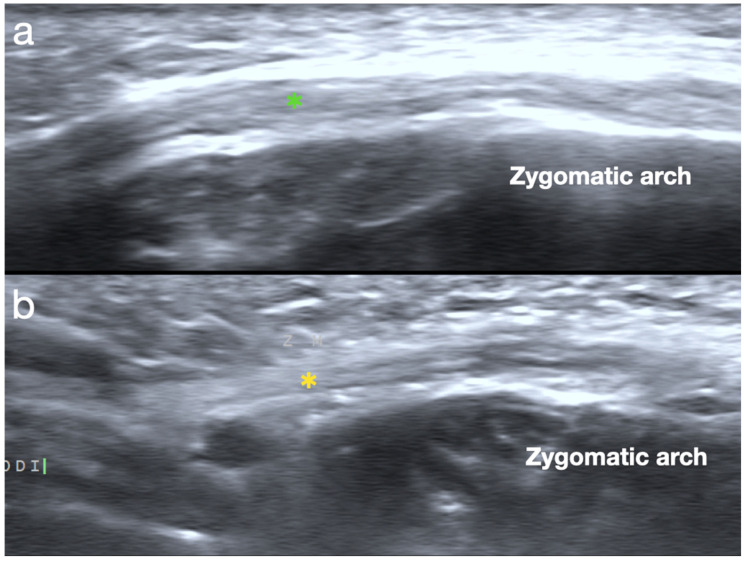
(**a**) Hypertrophied zygomatic major (ZMa) (green asterisk) muscle in wind player with no symptoms. (**b**) Grade II tear with hyperechoic change (yellow asterisk) inside of ZMa muscle with lack of tension during playing.

**Figure 8 diagnostics-15-00514-f008:**
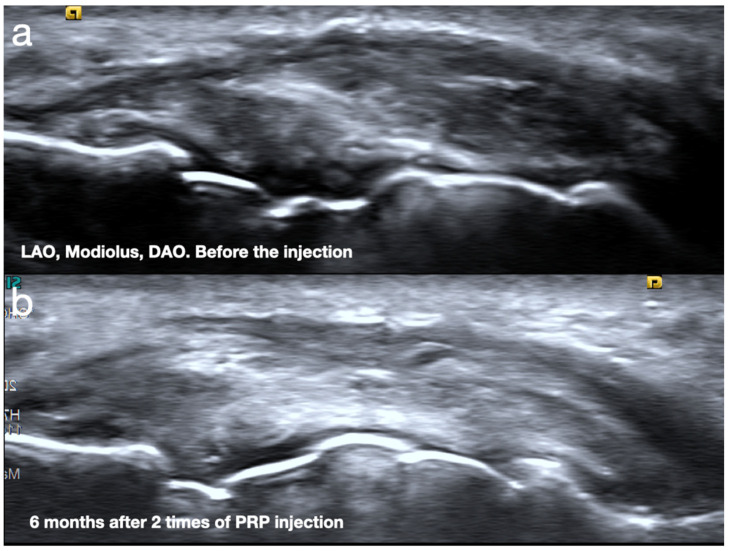
The healing of the injured embouchure 6 months after two platelet-rich plasma injections. (**a**) Before the injection, the partial tear is noted over the levator anguli oris (LAO), modiolus, and depressor anguli oris (DAO). (**b**) The healing of the LAO, modiolus, and DAO is reported 6 months after PRP injections, with the total recovery of embouchure function.

## Data Availability

No data available.

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
