# Peer review of "Novel Ultrasound Examination and Guided Intervention of Peri-Oral Musculature and Fascia in Wind Players with Embouchure Problems: Technical Note"

_diagnostics, 2025, doi:10.3390/diagnostics15050514_

Round 1
Reviewer 1 Report
Comments and Suggestions for Authors
Strengths
Innovative ultrasound-based approach to embouchure dysfunction.
Detailed sonoanatomy and scanning protocol.
Potential clinical impact on musicians' performance and treatment.
Key Concerns & Recommendations
Recognition of Affected Muscles
Define sonographic criteria for diagnosing strain, tears, and fibrosis.
Consider elastography for improved assessment.
Intervention Options
Discuss alternatives like dry needling and botulinum toxin.
Address efficacy of regenerative injections in different injury types.
Standardization & Functional Assessment
Ensure reproducibility across instruments and players.
Propose a standardized dynamic ultrasound protocol.
Comparison with Existing Methods
Contrast with EMG, MRI, and clinical assessments to strengthen rationale.
Long-Term Monitoring
Recommend follow-up ultrasound for recovery tracking.
Discuss prognosis based on intervention types.
Broader Applications
Potential relevance for speech therapy and maxillofacial rehab.
Clarifications Needed
Define terms like “failure of fascial tension” and “tensegrity.”
Specify recommended scanning frequency and depth.
Minor Comments
The manuscript would benefit from additional figures illustrating pathological ultrasound findings.
Clarify the recommended scanning frequency and depth for different muscle layers.
Consider discussing potential risks or limitations of ultrasound-guided interventions in this region.
Conclusion
The study is a valuable contribution, but refining diagnostic criteria, intervention options, and long-term monitoring will enhance clinical applicability.
Author Response
Strengths
Innovative ultrasound-based approach to embouchure dysfunction.
Detailed sonoanatomy and scanning protocol.
Potential clinical impact on musicians' performance and treatment.
Key Concerns & Recommendations
Recognition of Affected Muscles
Define sonographic criteria for diagnosing strain, tears, and fibrosis.
Response:
Thank you for your review and suggestions.
We have added, Lines 199 – 203:
We can classify the severity of muscle injury based on Chan et al. [12], Grade I is characterized by increased echogenicity without architectural distortion, Grade II is defined by discontinuous muscle fibers with altered echogenicity, and Grade III presents as complete discontinuity of muscle fibers with retraction of the muscle ends.
Consider elastography for improved assessment.
Response:
Thank you for your suggestion.
We have added the lines 314 – 316: This underscores the need for further research, including the addition of other scanning modalities, such as elastography, and their clinical application.
Intervention Options
Discuss alternatives like dry needling and botulinum toxin.
Response:
Thank you for your suggestion.
We have added lines 247 – 250: Literature suggests that dry needling is effective in reducing muscle stiffness and alleviating pain in muscle injuries [13]. However, when applied to wind players with embouchure dysfunction, the results are often unsatisfactory, particularly regarding embouchure control and durability while playing.
Address efficacy of regenerative injections in different injury types.
Response:
Thank you for your suggestion.
We have added lines 251 – 262: Regenerative injection therapy is known to promote tissue healing, and ultra-sound-guided injections of regenerative agents, such as high concentrations of dextrose or platelet-rich plasma, have been shown to facilitate tissue recovery [14-16].
According to a case series by Yeh et al., 19 wind players with embouchure injuries, with symptoms persisting for an average of 3.8 years, underwent platelet-rich plasma injections administered twice, two weeks apart. Among them, 81% reported improved embouchure control and a return to regular performance, with an average recovery time of four months. Those who did not respond well exhibited symptoms such as air leakage at the edge of the mouthpiece or dystonic movements of the embouchure or facial muscles, indicating severe embouchure dysfunction and/or embouchure dystonia [17]. In players who showed improvement, hyperechoic tissue with increased fiber continuity was observed (Figure 8).
Standardization & Functional Assessment
Ensure reproducibility across instruments and players.
Propose a standardized dynamic ultrasound protocol.
Response:
Thank you for your comment.
The “Scanning Protocol for the Embouchure in Wind Players” has been listed on lines 144 – 195:
Comparison with Existing Methods
Contrast with EMG, MRI, and clinical assessments to strengthen rationale.
Response:
Thank you for your suggestion.
We have added lines 66-70 and reference [7]:
Line 66 – 70: Current literature primarily utilizes surface electromyography to assess the function of embouchure muscles. However, distinguishing individual muscle activity remains challenging due to the close proximity of these muscles [7]. Additionally, there is a lack of literature exploring the use of ultrasound for diagnosing embouchure-related injuries.
Long-Term Monitoring
Recommend follow-up ultrasound for recovery tracking.
Discuss prognosis based on intervention types.
Response:
Thank you for your suggestion.
We have added the prognosis based on dry needling and PRP injections in lines 247 – 262.
Broader Applications
Potential relevance for speech therapy and maxillofacial rehab.
Response:
Thank you for your suggestion.
We have added lines 300 – 307: Ultrasound scanning of the perioral structures can be a valuable tool in speech therapy and maxillofacial rehabilitation. By providing real-time imaging of the muscle and soft tissue dynamics, practitioners can better assess the functional integrity of the embouchure and surrounding areas. This imaging allows for targeted interventions, enhancing the effectiveness of rehabilitation strategies. Furthermore, integrating ultrasound into therapy can help tailor individualized treatment plans, monitor progress, and adjust techniques based on visual feedback, ultimately improving outcomes for patients with embouchure dysfunction and related speech challenges.
Clarifications Needed
Define terms like “failure of fascial tension” and “tensegrity.”
Specify recommended scanning frequency and depth.
Response:
We have added reference [4, 5] to illustrate fascial tension detection through sonopalpation.
Minor Comments
The manuscript would benefit from additional figures illustrating pathological ultrasound findings.
Response: Thank you for your comment. We have included some examples of the sonopathologies. The most important consideration is comparing them with the normal side.
Lines 205 – 208: When the probe is moved sagittally to scan the origins of the LAO and DAO, an interruption with irregularity in the fibrillary pattern of the muscle and fascial layer indicates a Grade II tear. Additionally, comparing both sides of the embouchure can help differentiate abnormal findings from normal ones (Figure 6).
Clarify the recommended scanning frequency and depth for different muscle layers.
Response:
Thank you for your comments and suggestions.
We have added in the manuscript and together with the original information.
Lines 153 – 156: A linear transducer with a frequency of no less than 18 MHz is recommended, with the focus set at a depth of 1 to 2 cm. The depth may need to be adjusted to 2 to 3 cm deep for buccinator muscles.
Consider discussing potential risks or limitations of ultrasound-guided interventions in this region.
Response:
We have already listed in lines 290 – 298: Lastly, several neuromuscular structures should be visualized prior to intervention in the perioral embouchure to avoid puncture. First, the facial artery typically runs lateral to the modiolus, branching into the superior labial and inferior labial arteries, which supply the upper and lower part of the OO. Second, the buccal branch of the facial nerve runs above the buccinator and gives off several branches that form the infraorbital plexus. Since some of these nerves are too small to be adequately detected with ultrasound, the needle should be inserted slowly through the perioral region. Additionally, a hydrodissection technique [18] may be employed, using the injectate to push small blood vessels and nerves away, thereby minimize the risk of damage to these delicate structures.
Conclusion
The study is a valuable contribution, but refining diagnostic criteria, intervention options, and long-term monitoring will enhance clinical applicability.
Response:
Thank you for your review and the constructive suggestions. We appreciate your feedback. With this revision, we hope to have met the review requirements and enhanced the professionalism, impact, and clinical applications of the manuscript.
Reviewer 2 Report
Comments and Suggestions for Authors
Thank you for the opportunity to review the manuscript entitled „Novel Ultrasound Examination and Guided Intervention of Peri-oral Musculature and Fascia in Wind Players with Embouchure Problems: A Technical Note”
The authors of this technical note address a very interesting, yet niche issue of the occurrence of disorders in the perioral muscles and fascia in wind players.
The authors point out that this technical note present a scanning method for the wind players presented with embouchure problems, with common pathological sonographic findings, and ultrasound-guided interventions. The authors also emphasize the potential of ultrasound in diagnosing and treating problems contributing to effective therapeutic strategies for wind players.
Unfortunately, in my opinion, there are too few objective guidelines in this technical note and too many subjective observations. This unfortunately does not allow for practical use by other researchers. This diminishes the value of this work and should be corrected.
The authors have three goals for this technical note: to provide a comprehensive overview of the sonoanatomy of the peri-oral musculature relevant to embouchure; second, to present a scanning protocol for embouchure in wind players and to highlight common pathological sonographic findings associated with embouchure problems; and third, to detail ultrasound-guided interventions that may assist in the diagnosis and treatment of these issues.
If a detailed description of the anatomy was made, then I don't quite understand where the knowledge about typical sonographic pathologies comes from (whether based on empirical experience, literature review)? This needs to be written. If empirical, then based on how many cases of such disorders. If from literature, then it would be worth citing it!
The description regarding ultrasound-guided interventions is also too general when it comes to the third goal of this technical note.
Many paragraphs do not have any citations, which should be corrected.
Author Response
Reviewer 2:
Thank you for the opportunity to review the manuscript entitled “Novel Ultrasound Examination and Guided Intervention of Peri-oral Musculature and Fascia in Wind Players with Embouchure Problems: A Technical Note”
The authors of this technical note address a very interesting, yet niche issue of the occurrence of disorders in the perioral muscles and fascia in wind players.
The authors point out that this technical note present a scanning method for the wind players presented with embouchure problems, with common pathological sonographic findings, and ultrasound-guided interventions. The authors also emphasize the potential of ultrasound in diagnosing and treating problems contributing to effective therapeutic strategies for wind players.
Unfortunately, in my opinion, there are too few objective guidelines in this technical note and too many subjective observations. This unfortunately does not allow for practical use by other researchers. This diminishes the value of this work and should be corrected.
Response: Thank you for your time and effort in reviewing our manuscript. We appreciate your feedback and have made several changes to enhance the objectivity of the scanning methods discussed in the manuscript.
The authors have three goals for this technical note: to provide a comprehensive overview of the sonoanatomy of the peri-oral musculature relevant to embouchure; second, to present a scanning protocol for embouchure in wind players and to highlight common pathological sonographic findings associated with embouchure problems; and third, to detail ultrasound-guided interventions that may assist in the diagnosis and treatment of these issues.
If a detailed description of the anatomy was made, then I don't quite understand where the knowledge about typical sonographic pathologies comes from (whether based on empirical experience, literature review)? This needs to be written. If empirical, then based on how many cases of such disorders. If from literature, then it would be worth citing it!
Response: Thank you for your comment. We have added the following into the manuscript.
Line 66 – 70: Current literature primarily utilizes surface electromyography to assess the function of embouchure muscles. However, distinguishing individual muscle activity remains challenging due to the close proximity of these muscles [7]. Additionally, there is a lack of literature exploring the use of ultrasound for diagnosing embouchure-related inju-ries.
We have added the reference [7].
Line 155 – 156: The depth may need to be adjusted to 2 to 3 cm deep for buccinator muscles.
Lines 199 – 203: We can classify the severity of muscle injury based on Chan et al. [12], Grade I is char-acterized by increased echogenicity without architectural distortion, Grade II is defined by discontinuous muscle fibers with altered echogenicity, and Grade III presents as complete discontinuity of muscle fibers with retraction of the muscle ends.
We have added reference [12].
Lines 205 – 208: When the probe is moved sagittally to scan the origins of the LAO and DAO, an interruption with irregularity in the fibrillary pattern of the muscle and fascial layer indicates a Grade II tear. Additionally, comparing both sides of the embouchure can help differentiate abnormal findings from normal ones (Figure 6).
The description regarding ultrasound-guided interventions is also too general when it comes to the third goal of this technical note.
Response:
Thank you for your comment and insight, we have added the following in the manuscript.
Lines 247 – 262: Literature suggests that dry needling is effective in reducing muscle stiffness and alle-viating pain in muscle injuries [13]. However, when applied to wind players with em-bouchure dysfunction, the results are often unsatisfactory, particularly regarding em-bouchure control and durability while playing.
Regenerative injection therapy is known to promote tissue healing, and ultra-sound-guided injections of regenerative agents, such as high concentrations of dextrose or platelet-rich plasma, have been shown to facilitate tissue recovery [14-16].
According to a case series by Yeh et al., 19 wind players with embouchure injuries, with symptoms persisting for an average of 3.8 years, underwent platelet-rich plasma injections administered twice, two weeks apart. Among them, 81% reported improved embouchure control and a return to regular performance, with an average recovery time of four months. Those who did not respond well exhibited symptoms such as air leakage at the edge of the mouthpiece or dystonic movements of the embouchure or facial muscles, indicating severe embouchure dysfunction and/or embouchure dystonia [17]. In players who showed improvement, hyperechoic tissue with in-creased fiber continuity was observed (Figure 8).
We have also added lines 284 – 288: When performing ultrasound-guided injections to the embouchure structures, either an out-of-plane or in-plane approach can be used, with approximately 0.5 to 1 cc of injectate administered at each lesion. Some post-injection swelling may occur, but it is generally tolerable and subsides within a few days. Players should be advised to re-duce the intensity of their playing during the recovery period to facilitate optimal healing.
Many paragraphs do not have any citations, which should be corrected.
Response:
Thank you for your comments, we have added references accordingly as stated above.
Round 2
Reviewer 2 Report
Comments and Suggestions for Authors
Thank you for resubmitting the manuscript for review. The authors have made changes to the manuscript in accordance with the comments. The value of this manuscript has improved. I believe that in its current form it can be published in the journal.